# Cardiolipin-Based Lipopolyplex Platform for the Delivery of Diverse Nucleic Acids into Gram-Negative Bacteria

**DOI:** 10.3390/ph12020081

**Published:** 2019-05-28

**Authors:** Federico Perche, Tony Le Gall, Tristan Montier, Chantal Pichon, Jean-Marc Malinge

**Affiliations:** 1Centre de Biophysique Moléculaire, UPR4301 CNRS, Rue Charles Sadron, CEDEX 02, 45071 Orléans, France; federico.perche@cnrs-orleans.fr (F.P.); chantal.pichon@cnrs.fr (C.P.); 2Unité INSERM 1078, Faculté de Médecine, Université de Bretagne Occidentale, Université Européenne de Bretagne, 22 avenue Camille Desmoulins, CEDEX 3, 29238 Brest, France; tony.legall@univ-brest.fr (T.L.G.); tristan.montier@univ-brest.fr (T.M.)

**Keywords:** bacterial delivery, oligonucleotide, RNA, pDNA, ssDNA, dsDNA

## Abstract

Antibiotic resistance is a growing public health concern. Because only a few novel classes of antibiotics have been developed in the last 40 years, such as the class of oxazolidinones, new antibacterial strategies are urgently needed (Coates, A.R. et al., 2011). Nucleic acid-based antibiotics are a new type of antimicrobials. However, free nucleic acids cannot spontaneously cross the bacterial cell wall and membrane; consequently, their intracellular delivery into bacteria needs to be assisted. Here, we introduce an original lipopolyplex system named liposome polymer nucleic acid (LPN), capable of versatile nucleic acid delivery into bacteria. We characterized LPN formed with significant therapeutic nucleic acids: 11 nt antisense single-stranded (ss) DNA and double-stranded (ds) DNA of 15 and 95 base pairs (bp), 9 kbp plasmid DNA (pDNA), and 1000 nt ssRNA. All these complexes were efficiently internalized by two different bacterial species, i.e., *Escherichia coli* and *Pseudomonas aeruginosa*, as shown by flow cytometry. Consistent with intracellular delivery, LPN prepared with an antisense oligonucleotide and directed against an essential gene, induced specific and important bacterial growth inhibition likely leading to a bactericidal effect. Our findings indicate that LPN is a versatile platform for efficient delivery of diverse nucleic acids into Gram-negative bacteria.

## 1. Introduction

Antibiotic resistance is becoming a threat to global human health both in high- and low-income countries, resulting in a growing number of infections and increased mortality, especially due to Gram-negative bacteria [1,2,3]. The concept of using small nucleic acids as therapeutics to fight antibiotic resistant bacteria is very promising. Different types of strategies can be developed depending on the nature, form, and length of the nucleic acid to be delivered. Chemically synthesized ss- and dsDNA oligonucleotides can be used in order to target either mRNA (antisense strategy) [4] or transcription factor proteins (decoy strategy) [5], with the aim of specifically inhibiting critical bacterial gene expression. Aptamers and CRISPR-Cas nucleases have also recently emerged as potential antibacterial strategies [6,7,8]. Moreover, functional studies on non-coding bacterial small RNAs have recently highlighted their role in the regulation of genes associated with multiresistance [9] and opened the way to exploit ssRNA molecules as potential antimicrobial molecules [10]. RNA can be delivered or produced in bacteria by pDNA used as expression vector [11].

Because the cell wall of bacteria strongly impedes the penetration of any nucleic acids into the bacterial cytoplasm, a single versatile system suitable for efficient delivery of different types of nucleic acid would be particularly useful. To our knowledge, such a universal platform has not yet been reported.

Indeed, various delivery systems have been used, until now, to condense, protect, and carry nucleic acids into bacteria. Antisense oligodeoxynucleotides (ASOs) were mainly conjugated to cell-penetrating peptides (CPP) [4,12,13,14]. However, the CPP-dependent approach showed several limitations, including the requirement of designing derivatives when resistance to delivery arises from mutations occurring in bacterial transporter proteins [15]. Other strategies involve formulations with anionic liposomes for delivery of ASO [16] and DNA decoy oligonucleotide [17]. A more recent strategy uses lipopolyplexes with ASO being first complexed with polyethylenimine (PEI) before encapsulation of the PEI/ASO polyplex into anionic liposomes, yielding efficient delivery into Gram-negative bacteria [18,19,20,21].

Our group has developed a ternary delivery system named liposome polymer nucleic acid (LPN) for mRNA vaccine delivery [22,23,24,25]. LPN is relatively simple to prepare and stable in physiological fluids. We used them for the delivery of pDNA [26], mRNA [23,24,25], and siRNA [27] into mammalian cells. The two-step complexation protocol of LPN preparation results in the formation of cationic polymer/nucleic acid complexes (polyplexes) of similar size and charge—with different types of nucleic acids—before encapsulation into anionic liposomes, and is required to avoid inactivation of polyplexes in serum [28]. We mixed cationic polyplexes with anionic liposomes aiming to produce near-neutral nanoparticles for further in vivo antibacterial applications, as such nanoparticles are expected to show prolonged blood circulation and therefore higher delivery activity [29,30,31]. Indeed, neutral nanoparticles show negligible protein adsorption and decreased complement activation as compared to positive and negative ones [29,30,32].

Whereas the delivery of nucleic acids into bacteria has been developed with one formulation per nucleic acid type limiting the development of antibacterial nucleic acid therapeutics, we herein designed a novel LPN platform for the delivery of nucleic acids of diverse size (from 11 nt to 9514 bp) and nature (DNA or RNA) into bacteria.

We tested our LPN strategy on two model species of infectious bacteria, i.e., *Escherichia coli* and *Pseudomonas aeruginosa*. Most of antibiotic resistant infections worldwide are mainly due to *E. coli* [2]. *E. coli* is a Gram-negative bacteria responsible for causing severe foodborne infections which can lead to life-threatening hemolytic uremic syndrome [2]. Moreover, *E. coli* strains resistant to conventional antibiotics have an increasing prevalence in hospital-acquired infections [33,34], making *E. coli* a suitable model for nucleic acid delivery. *P. aeruginosa* is another serious cause of human healthcare-associated infections, with a high prevalence of multidrug-resistant strains and chronic infection in cystic fibrosis patients [35,36].

The present study reports the delivery of various forms of potential nucleic acid therapeutics into bacteria using novel LPNs as shown by flow cytometry. Moreover, antibacterial activity data support the therapeutic potential of LPN–ASO.

## 2. Results

### 2.1. Preparation and Characterization of LPN

Lipopolyplexes prepared with anionic liposomes composed of egg phosphatidylcholine (EPC)/dimyristoylphosphatidylglycerol (DMPG)/distearoyl-glycero-phosphoethanolamine-polyethyleneglycol (DSPE-PEG) liposomes encapsulating PEI polyplexes have been reported to deliver ASOs into bacteria [18,19,20,21]. Instead of using EPC/DMPG/DSPE-PEG liposomes to encapsulate the PEI polyplexes, we used a more simple mix of two components i.e., distearoyl-glycero-phosphocholine (DSPC) with the negatively charged phospholipid cardiolipin (CL) [37,38]. This choice was driven by the fact that CL interacts with the membrane of Gram-negative bacteria [39].

LPNs are especially suited for the versatile delivery of nucleic acids in eukaryotic cells [22]. Nucleic acids are first complexed with the established cationic polymer 25 kDa PEI, which has already been used in clinical trials [40], into polyplexes before mixing with anionic liposomes to form LPN (Figure 1).

Polyplexes size increased along with nucleic acid molecular weight: from 100–120 nm for 11nt ss ASO, 15 and 95 bp ds DNA to 166 nm for 1000 nt RNA and 214 nm for 9514 bp pDNA (Table 1). Entrapping the polyplexes increased the liposome diameter from 100 nm for liposome alone to 140–180 nm for LPN (Table 2) and resulted in near-neutral zeta potentials, physico-chemical features previously observed [24,41,42].

For DNA complexes, the size of LPN increased along with the molecular weight: 140/8 nm for ssASO DNA, 157/9 nm for 95 bp DNA oligonucleotide and 174/10 nm for 9514 bp pDNA. The size of RNA LPN was 156/15 nm, a value between that of 95 bp DNA and pDNA. Concerning the charge, all LPNs had a neutral to slightly positive charge (0.5–7 mV) (Table 2). Gel retardation electrophoresis confirmed the condensation of nucleic acids with no detectable free nucleic acid for ssASO DNA, 95 bp dsDNA, pDNA, and ssRNA (Figure 2A).

The morphology of LPN was determined by electron microscopy (Figure 2B) and was found in accordance with previous studies, with a dense polyplex surrounded by a liposome [42,43].

### 2.2. Delivery

Next, we evaluated nucleic acid delivery into bacteria using flow cytometry. In order to exclusively consider the intracellular signal of the FITC-labeled nucleic acids, extracellular fluorescence of the labeled nucleic acids into bacteria was quenched with trypan blue [23,44,45]. LPNs were able to deliver the different nucleic acids, irrespective of the size; indeed, ssASO DNA, 15 bp and 95 bp dsDNA, and 9514 bp pDNA were delivered into nearly 50% of *E. coli* and *P. aeruginosa* cells, with ASO delivery into cells having the highest efficacy (Figure 3). In both bacterial species, the percentage of transfected cells was lower with RNA, i.e., 15% and 25% in *E. coli* and *P. aeruginosa,* respectively.

### 2.3. Antibacterial Activity

To evaluate the efficiency of LPN in delivering ASO endowed with effect on bacteria growth inhibition, we delivered ASO targeting the acyl carrier protein (AcpP) mRNA whose sequence is well conserved between *E. coli* and *P. aeruginosa* species, with such mRNA targeting yielding effective growth inhibition as previously observed [14,46,47,48,49,50]. Delivery of 11 nt ssASO targeting AcpP mRNA reduced significantly the growth of *E. coli* (Figure 4A) and *P. aeruginosa* (Figure 4B). Such growth delay inhibition lasted at least 20 h (not shown). Growth decrease was no longer observed when LPNs were prepared with a single-stranded control ASO, confirming the specificity of AcpP mRNA targeting. In our experimental conditions, we also checked that cell viability was expectedly reduced when ampicillin was present in the growth medium.

### 2.4. Eukaryotic Safety

Next, we evaluated the toxicity of LPN–ASO towards mammalian cells. We observed that LPN–ASO was not toxic towards these eukaryotic cells with no loss in cell viability at the dose used on bacteria (Figure 5). Between 1and 4 µM ASO, we have observed that the absence of toxicity was unrelated to intracellular delivery (not shown). However, increasing the ASO concentration to 8-fold or more resulted in toxicity, with cell viability decreasing to less than 50%.

## 3. Discussion

Both antisense and decoy technologies are currently promising nucleic acid-based strategies for fighting against infectious diseases induced by antibiotic-resistant bacteria, although a main hurdle towards clinical development is still their weak cellular uptake [4,17,51,52]. In addition, our understanding of the role of ssRNA paves the way for using these RNA molecules as potential antibacterial therapeutics, and calling again for the design of an efficient delivery system. To enhance the efficacy of oligonucleotide therapeutics, various customized delivery systems have been designed to fit with the variable form and size of ASO and decoy oligonucleotides. Thus, there is a set of potential antibacterial nucleic acids but, no universal delivery system is available which would be very helpful in the search for new and efficient antimicrobial nucleic acids.

In this study, we report an original delivery system including CL, a component naturally found in the membrane of Gram-negative bacteria. CL is a dianionic tetra-acyl lipid featuring an overall conical shape. In Gram-negative bacteria, CL is a minor component of the outer membrane where it plays a pivotal role in its dynamic organization [53]. Its implication in the delivery of antimicrobial nanocomplexes in Gram-negative bacteria was demonstrated earlier [54,55]. CL could play the role of a helper in delivering nucleic acids [4], although the underlying mechanism has not yet been established. In the present study, we reasoned that CL could also be used as a main component of anionic liposomes employed to neutralize the positive charge of PEI/nucleic acid complexes within small-sized (<200 nm) nanocomplexes. This single LPN formulation is able to complex a wide range of nucleic acids—ssDNA ASO, dsDNA, pDNA, and RNA—into relatively small sizes (150–200 nm) nanoparticles. Results from flow cytometry indicate that LPNs are an efficient system to deliver this variety of nucleic acids into two relevant bacterial species, i.e., *E. coli* and *P. aeruginosa*.

Delivery of ASO in ≥50% of bacteria using LPN is higher than the 14% of *E. coli* cells transfected by free locked nucleic acids [56], and 30% of transfected cells with solid/lipid nanoparticles [17], but close to DNA delivery in 57% of bacteria, as previously reported with other anionic liposomes [16]. However, the uptake of LPN complexes into bacteria is still inferior to the 70% of *E. coli* cells positive for DNA after incubation with DNA nanopyramids, although the latter may interact with DNA binding sites on the bacterial membrane [57].

As for antibacterial potency, delivery of LPN–ASO targeting the essential gene *AcpP* showed the potential to specifically inhibit bacterial growth, almost as efficiently as an antibiotic. Complementary assays further demonstrated that this inhibition could lead to bacterial cell death (not shown). It is noteworthy here that these antibacterial effects were safe, specific and relevant, since they could be obtained for doses (of both the cargo and its vehicle) that were found to be fully safe for eukaryotic cells.

We observed that the uptake of 1000 nt RNA by both bacterial species was less important as compared to ASO and the different DNA species, although the extent of complexation of all nucleic acids considered was identical. It is plausible that unlike ssASO, which contains phosphorothioate backbone modifications for resistance to exonuclease activity, chemically unmodified ssRNA was partly digested outside and/or once inside bacteria, resulting in lower intracellular fluorescence. Both dsDNA duplex of 15 and 95 bp (which possessed three phosphorothioate linkages at each DNA end) showed identical delivery efficacy. This observation is of great interest in the search for efficient decoys as the length of consensus sequence recognized by bacterial targeted transcription factors is variable, and depends on the neighboring nucleotide sequence. In addition, the formulation reported herein could also be used to deliver circular DNA ds duplex (DNA minicircle), designed recently to trap several transcription factors for increased decoy strategy efficacy [58].

## 4. Materials and Methods

### 4.1. Reagents

All reagents were purchased from Sigma-Aldrich (St. Quentin Fallavier, France) unless otherwise stated. DSPC (1,2-distearoyl-*sn*-glycero-3-phosphocholine), DMPG (1,2-dimyristoyl-*sn*-glycero-3-phospho-(1′-rac-glycerol), CL (cardiolipin), NBD-PE (1,2-dioleoyl-*sn*-glycero-3-phosphoethanolamine-*N*-(7-nitro-2-1,3-benzoxadiazol-4-yl) (ammonium salt), Rhod-PE (1,2-dioleoyl-*sn*-glycero-3-phosphoethanolamine-*N*-(lissamine rhodamine B sulfonyl), were from Avanti Polar Lipids (Alabaster, AL, USA). Chemical structures of polymer and lipids used are provided in Figure A1.

### 4.2. Bacteria

*E. coli DH5α* (ATCC^®^ 53868) was obtained from Thermo Fisher (Les Ulis, France). *P. aeruginosa* (ATCC^®^ 10145) was obtained from the Pasteur Institute (Paris, France).

### 4.3. Nucleic Acids

ASOs with a phosphorothioate backbone were synthetized by Eurogentec (Angers, France). The sequence of the two ASOs was EC AcpP: C*T*T*C*G*A*T*A*G*T*G against *E. coli* [15] and PA AcpP: C*T*T*C*G*A*T*G*G*T*G against *P. aeruginosa* [59], which are complementary to the acyl carrier protein mRNA gene (bases 6–16 of the mRNA coding region) [46]. A control mismatch designated as control ASO (TCTCAGATGGT) was also used [15]. The pTG11033 pDNA (9514 bp) was a gift from Transgène S.A., Strasbourg, France [23]. Luciferase mRNA (1000 nt) was produced by in vitro transcription as in [23]. Nucleic acids were labeled with FITC (Fluorescein Isothiocyanate) using a commercial kit (Label IT^®^ Nucleic Acid Labeling Kits, Mirus Bio, Madison, WI, USA).

### 4.4. Liposome Preparation

Liposomes were prepared by film hydration as in [60]. Liposomes contained 42% CL and 58% DSPC (molar percentages). Chloroform solutions of lipids were evaporated in a rotary evaporator to form lipid films. The latter were hydrated with 10 mM pH 7.4 HEPES at a final lipid concentration of 5.4 mM. Liposomes were then sonicated for 15 min at 20 °C at 37 kHz using an ultrasonic bath (Fischer Bioblock Scientific, Illkirch, France).

### 4.5. Preparation of Lipopolyplexes

Lipopolyplexes were prepared according to a procedure modified from [23,25]. Instead of mixing polyplexes with a dried lipid film followed by sonication and extrusion, we mixed polyplexes with liposomes in HEPES solution. Briefly, to prepare lipopolyplexes, nucleic acids were mixed with branched PEI of 25kDa in HEPES 10 mM pH 7.4 at a weight ratio (PEI/nucleic acid) of 6 and at room temperature for 20 min, to allow for polyplex formation. Note that PEI solution was added to nucleic acid solution (and not the opposite), as the order of addition is important for LPN formation [61]. Liposomes were then added to polyplexes using 2 µL of liposome suspension per µg of nucleic acid.

### 4.6. LPN Characterizations

Size and zeta potential of LPNs were determined by DLS using an SZ-100 nanoparticle analyzer (Horiba, Longjumeau, France). LPNs were diluted in 10 mM pH 7.4 HEPES 40 mM NaCl and diameter and zeta potential were measured at 25 °C. Complexation of nucleic acids was validated by gel shift assays: 2 µg of free or LPN-complexed nucleic acids were run on 1% agarose-formaldehyde gels containing ethidium bromide. Gels were imaged using a Gene Flash imager (Syngene, Cambridge, UK). Transmission electronic microscopy samples were prepared according to the technique of negative staining using uranyl acetate. To this end, 5 µL of LPN solution in HEPES buffer was deposited on a carbon-coated copper grid for 5 min, and then adsorbed with filter paper. Then, 5 µL of uranyl acetate 2% in endonuclease-free water was deposited on the grid for 10 s and then adsorbed. Samples were dried at room temperature for 20 min before TEM observation. LPN structure was analyzed using a Philips CM20/STEM electron microscope operating at 50 kV (Centre de Microscopie Electronique, Université d’Orléans, France).

### 4.7. Flow Cytometry

Cell suspensions were analyzed with a FORTESSA X20 flow cytometer (Beckton Dickinson, Franklin Lakes, NJ, USA). The extracellular fluorescence of the NBD-PE lipid present in the liposomes or the FITC conjugated to the nucleic acid was quenched using trypan blue as in [23,44,45]. The cell-associated fluorescence was then measured with a flow cytometer (FACSort; Becton Dickinson, Franklin Lakes, New Jersey) with λex = 488 nm; λem = 530 ± 30 nm. The fluorescence intensity was expressed as the mean fluorescence intensity of 10,000 events.

### 4.8. Bacterial Growth Assay

Single-colony overnight cultures were diluted to 5 10^6^ cfu/mL in LB broth. Aliquots (100 µL) of this culture were transferred to 96-well plates. Aliquots (100 µL) of treatments in LB broth were then added to the wells at a final ASO concentration of 1 µM. Plates were grown at 37 °C with shaking, and bacterial growth was monitored by absorbance measurements at 600 nm every 15 min using a Victor I spectrophotometer (1420 Multilabel Counter Victor, Wallac, PerkinElmer, Courtaboeuf, France). As a control for growth inhibition, bacteria were treated with ampicillin at 150 µg/mL.

### 4.9. Cytotoxicity Towards Eukaryotic Cells

DC2.4 murine immortalized DC cells were a gift from Kenneth L. Rock [62], and were grown at 37 °C in a humidified atmosphere containing 5% CO_2_ in RPMI1640 medium supplemented with 10% heat-inactivated fetal bovine serum, 100 units/mL of penicillin and 100 μg/mL of streptomycin (Fischer Bioblock, Illkirch, France). Cells were mycoplasma-free as evidenced by MycoAlert Mycoplasma Detection Kit (Lonza, Levallois Perret, France). Cytotoxicity was evaluated performing an MTT (3-(4,5-dimethylthiazol-2-yl)-2,5-diphenyltetrazolium bromide) assay as in [23].

## 5. Conclusions

Our results indicate that LPN is a safe multipotent system efficient to deliver a wide range of nucleic acids into Gram-negative bacteria. Indeed, we have delivered DNA and RNA molecules of varying size and nature: RNA (1000 nt), pDNA (9514 bp), linear dsDNA (15 bp and 95 bp), and ASO (11 nt). This broad spectrum activity supports the potential of the LPN platform to treat bacterial infection by delivery of specific ASO or small RNA in different bacterial species.

## Figures and Tables

**Figure 1 pharmaceuticals-12-00081-f001:**
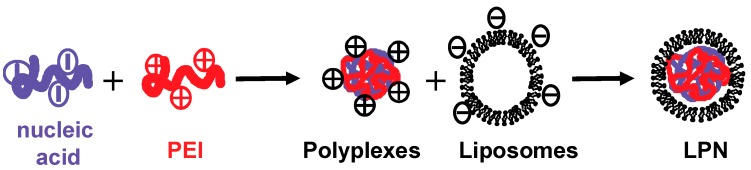
Two-step preparation of liposome polymer nucleic acid (LPN).

**Figure 2 pharmaceuticals-12-00081-f002:**
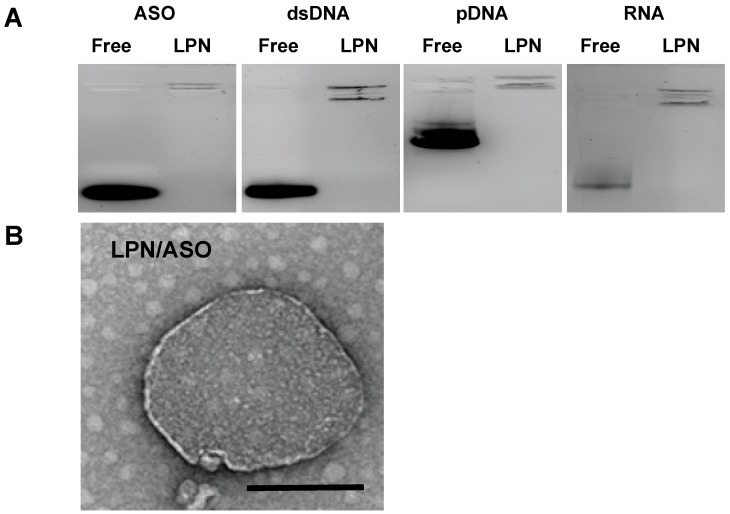
Characterization of LPN: (**a**) Gel retardation of nucleic acids either free or LPN-complexed. (**b**) Morphology of LPN–ASO complexes imaged by electron microscopy, scale bar represents 100 nm.

**Figure 3 pharmaceuticals-12-00081-f003:**
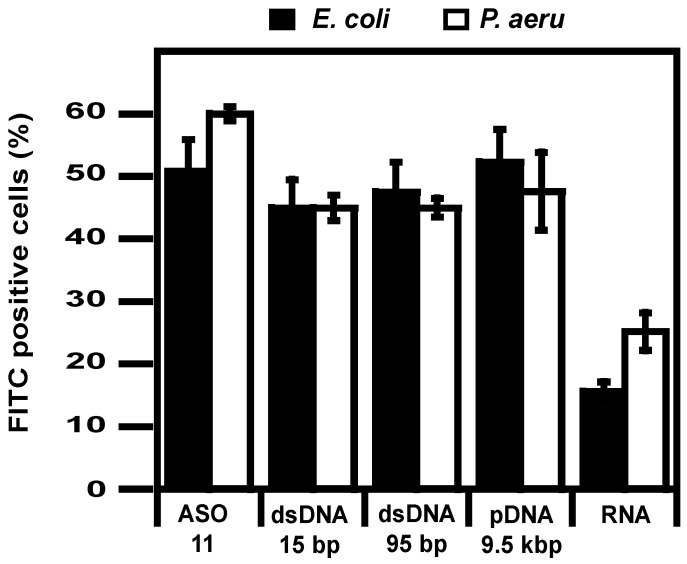
Nucleic acid delivery into bacteria. Delivery of LPN prepared with FITC-labeled nucleic acids was evaluated by flow cytometry after 2 h incubation at 37 °C with bacteria.

**Figure 4 pharmaceuticals-12-00081-f004:**
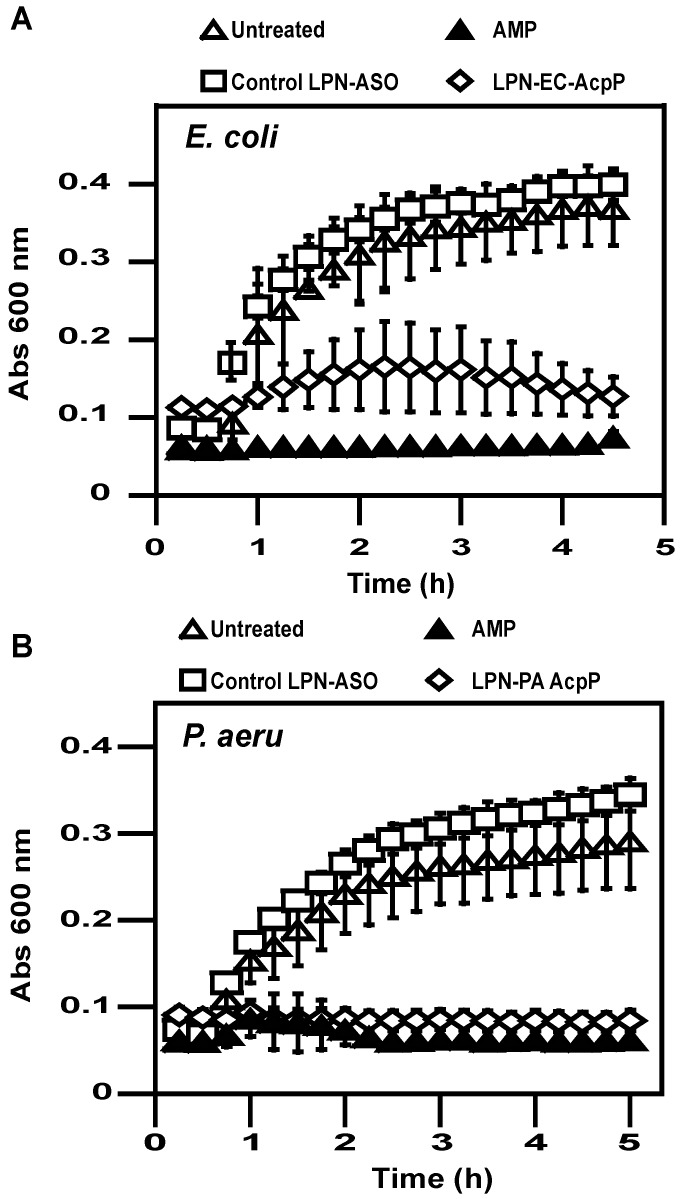
Antibacterial activity of LPN. The growth of *E. coli* (**a**) and *P. aeruginosa* (**b**) after treatment with LPN–ASO complexes (1 µM ASO) or ampicillin (AMP, 150 µg/mL) was monitored for 5 h at 37 °C.

**Figure 5 pharmaceuticals-12-00081-f005:**
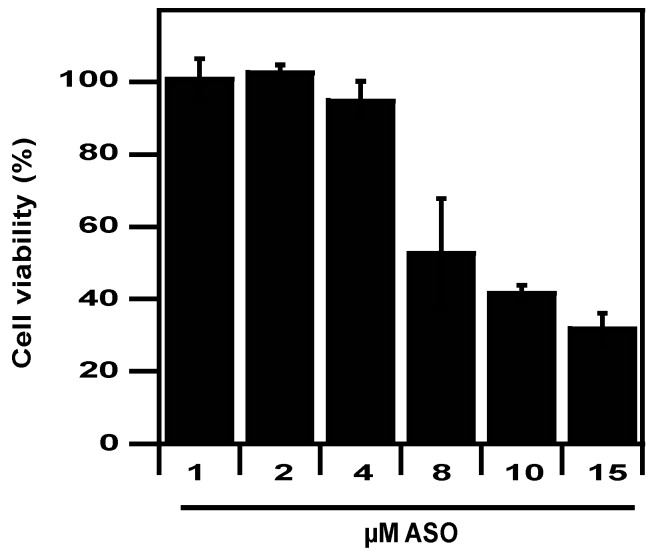
Cytotoxicity of LPN towards eukaryotic cells. Cell viability of dendritic cells 24 h after transfection with LPN complexes using increasing concentrations of ASO.

**Table 1 pharmaceuticals-12-00081-t001:** Size and zeta potential of polyethylenimine (PEI) polyplexes.

Nucleic Acid	Size (nm)	Zeta Potential (mV)
11 nt ssASO	100 ± 4	8 ± 1.5
15 bp dsDNA	120 ± 8	10 ± 3
95 bp dsDNA	100 ± 7	6.2 ± 1.7
9 kbp pDNA	214 ± 18	9 ± 4
1000 nt RNA	166 ± 10	13 ± 2

**Table 2 pharmaceuticals-12-00081-t002:** Size and zeta potential of LPN.

Nucleic Acid	Size (nm)	PDI	Zeta Potential (mV)
11 nt ssASO	140 ± 8	0.20 ± 0.02	6.1 ± 4.7
15 bp dsDNA	150 ± 12	0.26 ± 0.04	5 ± 3
95 bp dsDNA	157 ± 9	0.24 ± 0.04	9.7 ± 4.7
9 kbp pDNA	174 ± 10	0.40 ± 0.08	0.5 ± 0.4
1000 nt RNA	156 ± 15	0.27 ± 0.03	7.4 ± 3.7

PDI: polydispersity index.

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
