# Peer review of "Cardiolipin-Based Lipopolyplex Platform for the Delivery of Diverse Nucleic Acids into Gram-Negative Bacteria"

_pharmaceuticals, 2019, doi:10.3390/ph12020081_

Round 1

Reviewer 1 Report

The manuscript describes an interesting and straight forward set of experiments aiming to test the efficacy of a cardiolipin-based lipopolyplex for nucleic acids delivery within bacterial. Authors characterize the usefulness of this composition to deliver nucleic acids varying in nature and size, further they report about the inhibition of bacterial growth exerted by the delivery of nucleic acids targeting an essential mRNA.   

The work is technically and scientifically sound.

Minor points

1.- Both in the Abstract (line 18) and the discussion (line 152) authors state that the therapeutic classes of nucleic acids they used (i.e. antisense DNA and decoys) are the most advanced or significant therapeutic ones. This statement requires to be supported by references.

In fact this statement is questionable; there are other kinds of therapeutic nucleic acids that are currently gaining a lot of interest, like Aptamers (DNA or RNA), siRNA and nucleic acids coupled to the CRISPR-cas system. Authors should provide appropriate references and/or rephrasing such statements (Abstract and Discussion), they actually do not add any extra value to the work presented.

2.- It should be interesting to include in the Discussion section a paragraph enumerating the advantages of the cardiolipin-based compounds in comparison to others previously reported (e.g. Delivery efficiency, size of complexes, etc).

3.-  It should be interesting if the authors comment on the effect on bacterial growth at larger time points.

4.- (Line 63) What does LPD stand for?. Should it be LPN instead. LPN and LPD are the same or they refer to different complexes. Heading of table 2 says LPN but when describing in the text the results included in such Table lines 98-100 authors used LPD to refer to them. It is really confusing.

Review the text from line 63 to the end of the manuscript.

5.- Table 1, is it correct that the size of complexes with 15 bp dsDNA is larger than the one with 95 bp?

6.- Line 101, Table 2 instead of Table 1

7.- line 110, a space before it should be added

8.- No space should be included between a number and the % symbol (N%)

9.- lines 245 and 247 …?

10.- The formatting of the references list is not correct. Numbering appearing twice.

Author Response

The manuscript describes an interesting and straight forward set of experiments aiming to test the efficacy of a cardiolipin-based lipopolyplex for nucleic acids delivery within bacterial. Authors characterize the usefulness of this composition to deliver nucleic acids varying in nature and size, further they report about the inhibition of bacterial growth exerted by the delivery of nucleic acids targeting an essential mRNA.   

The work is technically and scientifically sound.

Minor points

1.- Both in the Abstract (line 18) and the discussion (line 152) authors state that the therapeutic classes of nucleic acids they used (i.e. antisense DNA and decoys) are the most advanced or significant therapeutic ones. This statement requires to be supported by references.

In fact this statement is questionable; there are other kinds of therapeutic nucleic acids that are currently gaining a lot of interest, like Aptamers (DNA or RNA), siRNA and nucleic acids coupled to the CRISPR-cas system. Authors should provide appropriate references and/or rephrasing such statements (Abstract and Discussion), they actually do not add any extra value to the work presented.

Response:

We agree with the first remark and we have modified two sentences in the abstract and in the beginning of the discussion (lines 18 and 161). As requested, we added references in the discussion to support the fact that these strategies are important nucleic acids based approaches developed actually to kill bacteria (line 163). 

We also agree that other therapeutic nucleic acids could be promising such as aptamers, siRNA and the crisp-cas system. As siRNA needs cellular machinery which is only present in eukaryotic cells, such interference RNA strategy cannot be used unfortunately in prokaryotes to our knowledge.  As requested, a sentence was added in the introduction to indicate that aptamers and Crisp cas nucleases system are promising nucleic acids based strategies including new references (lines 37-38).

In the Abstract, in the sentence “We characterized LPN formed with representative of most significant therapeutic nucleic acids: 11 nt antisense single-stranded (ss) DNA and double-stranded (ds) DNA of 15 and 95 base pairs (bp), 9 kbp plasmid DNA (pDNA) and 1,000 nt ss RNA.”, the term “most” has been removed.

In the Introduction, we added the following sentence: “Aptamers and CRISPR-Cas nucleases have also recently emerged as potential antibacterial strategies.[7-9]”

Three references were added:

1.         Bikard, D.; Barrangou, R., Using CRISPR-Cas systems as antimicrobials. Current opinion in Microbiology 2017, 37, 155-160.

2.         Hong, K. L.; Sooter, L. J., Single-stranded DNA aptamers against pathogens and toxins: Identification and biosensing applications. BioMed research international 2015, 2015.

3.         Davydova, A.; Vorobjeva, M.; Pyshnyi, D.; Altman, S.; Vlassov, V.; Venyaminova, A., Aptamers against pathogenic microorganisms. Critical reviews in microbiology 2016, 42 (6), 847-865.

We added 3 references in the Discussion:

1.         McArthur, M., Transcription factor decoys for the treatment and prevention of infections caused by bacteria including clostridium difficile. US Patent 9,024,005: 2014.

2.         McArthur, M., Transcription factor decoys. US Patent 9,550,991: 2017.

3.         González-Paredes, A.; Sitia, L.; Ruyra, A.; Morris, C. J.; Wheeler, G. N.; McArthur, M.; Gasco, P., Solid lipid nanoparticles for the delivery of anti-microbial oligonucleotides. European Journal of Pharmaceutics and Biopharmaceutics 2019, 134, 166-177.

4.         Xue, X.-Y.; Mao, X.-G.; Zhou, Y.; Chen, Z.; Hu, Y.; Hou, Z.; Li, M.-K.; Meng, J.-R.; Luo, X.-X., Advances in the delivery of antisense oligonucleotides for combating bacterial infectious diseases. Nanomedicine: Nanotechnology, Biology and Medicine 2018, 14 (3), 745-758.

2.- It should be interesting to include in the Discussion section a paragraph enumerating the advantages of the cardiolipin-based compounds in comparison to others previously reported (e.g. Delivery efficiency, size of complexes, etc).

Response: We added the following sentence in the Discussion: “In this study, we report an original delivery system including CL, a component naturally found in the membrane of Gram-negative bacteria. CL is a dianionic tetra-acyl lipid featuring an overall conical shape. In Gram-negative bacteria, CL is a minor component of the outer membrane where it plays a pivotal role in its dynamic organization.[54] Its implication in the delivery of antimicrobial nanocomplexes in Gram-negative bacteria was demonstrated earlier.[53]. CL could play the role of an helper in delivering nucleic acids [4] although the underlying mechanism is not yet established. In the present study, we reasoned that CL could also be used as a main component of anionic liposomes employed to neutralize the positive charge of PEI/nucleic acid complexes within small size (< 200 nm) nanocomplexes. “

The following references were added:

Renner, L. D.; Weibel, D. B., Cardiolipin microdomains localize to negatively curved regions of Escherichia coli membranes. Proceedings of the National Academy of Sciences 2011, 108 (15), 6264-6269.

Hegarty, J. P.; Krzeminski, J.; Sharma, A. K.; Guzman-Villanueva, D.; Weissig, V.; Stewart Sr, D. B., Bolaamphiphile-based nanocomplex delivery of phosphorothioate gapmer antisense oligonucleotides as a treatment for Clostridium difficile. International journal of nanomedicine 2016, 11, 3607.

3.-  It should be interesting if the authors comment on the effect on bacterial growth at larger time points.

Response: We added the sentence “Such growth inhibition lasted at least 20 hours (data not shown)”.

4.- (Line 63) What does LPD stand for?. Should it be LPN instead. LPN and LPD are the same or they refer to different complexes. Heading of table 2 says LPN but when describing in the text the results included in such Table lines 98-100 authors used LPD to refer to them. It is really confusing.

Review the text from line 63 to the end of the manuscript.

Response: We agree with the Reviewer and edited ”LPD” to “LPN” throughout the manuscript.

5.- Table 1, is it correct that the size of complexes with 15 bp dsDNA is larger than the one with 95 bp?

Response: Sizes of both complexes are similar: 120 +/- 8 nm for 15 bp DNA and 100 +/- 7 nm for 95 bp DNA, the difference is not significant.

6.- Line 101, Table 2 instead of Table 1

Response; Thank you for your comment, we made the change.

7.- line 110, a space before it should be added

Response; Thank you for your comment, we made the change.

8.- No space should be included between a number and the % symbol (N%)

Response; Thank you for your comment, we made the change throughout the text.

9.- lines 245 and 247 …?

Response: We are sorry and made the corrections.

10.- The formatting of the references list is not correct. Numbering appearing twice.

Response: We are sorry for this mistake which has been corrected in the revised version.

Reviewer 2 Report

The manuscript by Perche et al. reports the formulation of liposome-polymer-nucleic acid (LPN) nanoparticles for facilitating the bacterial uptake of nucleic acids and antibacterial activities. The authors employed gel shift, material characterization techniques such as TEM, DLS, flow cytometry, and cell culture studies to demonstrate that anionic liposome coating allows for selective targeting of nucleic acids towards bacteria over mammalian cells. The manuscript may be publishable after the below concerns are addressed.

Line 2, specify the two novel classes of antibiotics

Line 19, remove the word “oligonucleotides”

Lines 58-59, please explain why anionic liposome is required to avoid inactivation…

Line 63, 98, and 116, LPD should be LPN?

Lines 77-87, To demonstrate the advantage of the LPN formulation, the authors can show the chemical structures of the polymers and lipids in the Supporting Information.

Tables 1 and 2, Please indicate that “/” is equal to “+/-“. Could the authors explain why the sizes of LPN are larger than those of PEI polyplexes for 9 kbp pDNA and 1000-nt RNA?

Line 101, Table 1 should be Table 2.

Line 116 and Figure 3, Please explain how the large RNAs and DNAs are labelled with FITC. In Figure 3, the authors could also add controls of free RNAs and DNAs.

Figure 3, Could the authors add the flow cytometry data for mammalian cells for a direct comparison?

Figure 4, panel A, LPN-ACP1 should be LPN-EC-ACP1?

Lines 204-205, Please show the sequence of the control.

Line 223, add more details (buffers and temperatures, gel imaging method, etc) for DLS and gel experiments.

Lines 240, 245, 247, add the missing information.

Author Response

The manuscript by Perche et al. reports the formulation of liposome-polymer-nucleic acid (LPN) nanoparticles for facilitating the bacterial uptake of nucleic acids and antibacterial activities. The authors employed gel shift, material characterization techniques such as TEM, DLS, flow cytometry, and cell culture studies to demonstrate that anionic liposome coating allows for selective targeting of nucleic acids towards bacteria over mammalian cells. The manuscript may be publishable after the below concerns are addressed.

 Line 2, specify the two novel classes of antibiotics

Response: For safe of clarity we changed the sentence and we specified antibiotics class: oxazolidinones: “Antibiotic resistance is a growing public health concern. Since only few novel classes of antibiotics have been developed in the last 40 years such as the class of oxazolidinones, new antibacterial strategies are urgently needed.[1]”.

 Line 19, remove the word “oligonucleotides”

Response: We removed the word oligonucleotides.

 Lines 58-59, please explain why anionic liposome is required to avoid inactivation.

Response: We added the following sentences: “We mixed cationic polyplexes with anionic liposomes aiming to produce near-neutral nanoparticles for further in vivo antibacterial applications as such nanoparticles are expected to show prolonged blood circulation and therefore higher delivery activity.[29-31] Indeed, neutral nanoparticles show negligible protein adsorption and, decreased complement activation as compared to positive and negative ones.[29,30,32]”

The following references were added:

Ranneh, AH et al. (2018). An Ethylenediaminebased Switch to Render the Polyzwitterion Cationic at Tumorous pH for Effective Tumor Accumulation of Coated Nanomaterials. Angew Chem 130: 5151-5155.  

Chen, Let al. (2017). Effects of surface charge of hyperbranched polymers on cytotoxicity, dynamic cellular uptake and localization, hemotoxicity, and pharmacokinetics in mice. Mol Pharm 14: 4485-4497.

Meng, H et al. (2018). Walking the line: the fate of nanomaterials at biological barriers. Biomaterials. 2018, 174, 41-53.

Chonn, A et al (1991), The role of surface charge in the activation of the classical and alternative pathways of complement by liposomes. The Journal of immunology, 146 (12), 4234-4241.

 Line 63, 98, and 116, LPD should be LPN?

Response: We agree with the Reviewer and edited ”LPD” to “LPN” throughout the manuscript.

Lines 77-87, To demonstrate the advantage of the LPN formulation, the authors can show the chemical structures of the polymers and lipids in the Supporting Information.

Response: We added the structures of polymer and lipids in Supplementary Figure 1. No cell-associated fluorescence was detected with free nucleic acids.

 Tables 1 and 2, Please indicate that “/” is equal to “+/-“. Could the authors explain why the sizes of LPN are larger than those of PEI polyplexes for 9 kbp pDNA and 1000-nt RNA?

Response:

We thank the Reviewer for this comment, we changed “:” to “+/-“in Tables 1 and 2.

We edited the sentence “Mixing these positively-charged polyplexes with liposomes resulted in the formation of LPNs with a size of 140-180 nm and almost neutral potential (Table 2) which is in agreement with previous observations.[22,38,39]” to “Entrapping the polyplexes increased the liposome diameter from 100 nm for liposome alone to 140-180 nm for LPN (Table 2) and resulted in near neutral zeta potentials, physico-chemical features previously observed.[24,41,42]”

The major background arises from intrinsic cell fluorescence and not from nucleic acid material that unfortunately cannot cross the bacterial membranes.

 Line 101, Table 1 should be Table 2.

Response: We made the change.

Line 116 and Figure 3, Please explain how the large RNAs and DNAs are labelled with FITC. In Figure 3, the authors could also add controls of free RNAs and DNAs.

Response: We added this information:” Nucleic acids were labeled wih FITC using a commercial kit (Label IT® Nucleic Acid Labeling Kits, Mirus Bio, Madison, WI, USA).”

 Figure 3, Could the authors add the flow cytometry data for mammalian cells for a direct comparison?

Response: We edited the sentence “We observed that LPN-ASO was not toxic towards these eukaryotic cells with no loss in cell viability at the dose used on bacteria showing selectivity of our formulation for bacterial activity (Fig. 5).” To “We observed that LPN-ASO was not toxic towards these eukaryotic cells with no loss in cell viability at the dose used on bacteria Fig. 5).” We also added the following sentence: “From 1 to 4 µM in ASO concentration, we have observed that the absence of toxicity was unrelated to intracellular delivery (data not shown).

 Figure 4, panel A, LPN-ACP1 should be LPN-EC-ACP1?

Response: Thank you for the comment, we made the change in the figure.

 Lines 204-205, Please show the sequence of the control.

Response: We added the control sequence (TCTCAGATGGT) which is from Pukett et al. as referenced

 Line 223, add more details (buffers and temperatures, gel imaging method, etc) for DLS and gel experiments.

Response: We added more details on these methods: “LPN were diluted in 10 mM pH 7.4 Hepes 40 mM NaCl and both diameter and zeta potential were measured at 25°C. Complexation of nucleic acids was validated by gel shift assays: 2µg of free or LPN-complexed nucleic acids were run on 1% agarose-formaldehyde gels containing ethidium bromide. Gels were imaged using a Gene Flash imager (Syngene, Cambridge, UK).”

 Lines 240, 245, 247, add the missing information.

Response: We added the missing information: line 240 (5X106 CFU/mL), line 245 (150 µg/mL), line 247 (immortalized DC).

Round 2

Reviewer 2 Report

The revised manuscript is acceptable for publication.